Elevated SLC1A5 links to inflamed endothelial cells and proteinuria in membranous nephropathy patients

Tang Qingqin 1
Ling Yi 2
Li Jianzhong 2
Feng Bin 1
Zhang Sheng 1
Xu Deyu 2
Zhou Ling 2
Shen Lei 2
Lu Guoyuan 2
Chen Mingyu 2
Qiao Longwei qiaolongwei1@126.com 3
Liang Yuting liangyuting666@126.com 1 4
1 Center for Clinical Laboratory, The First Affiliated Hospital of Soochow University , Suzhou, Jiangsu Province , China
2 Department of Nephrology, The First Affiliated Hospital of Soochow University , Suzhou, Jiangsu Province , China
3 Center for Reproduction and Genetics, School of Gusu, The Affiliated Suzhou Hospital of Nanjing Medical University, Suzhou Municipal Hospital, Nanjing Medical University , Suzhou, Jiangsu Province , China
4 Molecular Oncology Laboratory, Department of Orthopaedic Surgery and Rehabilitation , Chicago, Illinois , United States of America
Anson Lesley
Electronic publication date: 2025 Oct 31
Publication date: 2025
Volume: 13
Electronic Location ID: e20271
Received 2024 Dec 2; Accepted 2025 Sep 30
Copyright: ©2025 Tang et al.
Copyright year: 2025
Copyright holder: Tang et al.
License: This is an open access article distributed under the terms of the Creative Commons Attribution License, which permits unrestricted use, distribution, reproduction and adaptation in any medium and for any purpose provided that it is properly attributed. For attribution, the original author(s), title, publication source (PeerJ) and either DOI or URL of the article must be cited.
License URL: https://creativecommons.org/licenses/by/4.0/

Keywords: Membranous nephropathy, SLC1A5, Single cell, Mendelian randomization, Proteinuria

Funding: Science Foundation of Jiangsu Province No. BK2023119 No. BK20240371 Science and Technology Project of Suzhou No. SKY2022141 National Science Foundation of China No. 81700589 No. 81901632 This work was supported by the Science Foundation of Jiangsu Province Grant (No. BK2023119 and No. BK20240371); Science and Technology Project of Suzhou (No. SKY2022141); National Science Foundation of China Grants (No. 81700589 and No. 81901632). The funders had no role in study design, data collection and analysis, decision to publish, or preparation of the manuscript.

==============================
Background

Membranous nephropathy (MN) is an autoimmune glomerular disorder characterized by persistent proteinuria. Elucidating its pathophysiological mechanisms and signaling pathways is crucial for improving diagnostic and therapeutic strategies.

Methods

We performed differential analysis on the MN glomerular transcriptome and assessed immune infiltration. Single-cell analysis identified key genes’ subcellular localization, while pseudotime and cell communication analyses determined subpopulations linked to MN progression. Genes causally related to MN onset were screened using Mendelian randomization, and serum core genes were correlated with proteinuria via ELISA.

Results

Transcriptome analysis revealed 95 differentially expressed genes, predominantly enriched in immune and metabolic pathways. Macrophage polarization played a pivotal role in MN, with monocytes/macrophages and endothelial cells identified as key contributors. Pseudotemporal analysis showed elevated pro-inflammatory macrophages and inflammatory endothelial cells in high-proteinuria patients. Macrophage-endothelial cell communication involved key signaling molecules. hdWGCNA analysis identified three molecular sets linked to inflammatory cells, with Mendelian randomization confirming their causal relationship to MN. SLC1A5 was identified as a key gene, and serum sample validation confirmed its strong correlation with proteinuria.

Conclusion

This study identified novel macrophage and endothelial cell subtypes and their interactions, positioning SLC1A5 as a potential biomarker for MN pathogenesis.

Introduction

Membranous nephropathy (MN) is an autoimmune glomerular disease affecting the kidneys (Cai et al., 2023; Liu et al., 2022). Its pathogenesis involves circulating antibodies targeting podocytes, which deposit beneath the glomerular epithelium to form immune complexes. This process results in podocyte damage, glomerular basement membrane thickening, and proteinuria, establishing MN as a leading cause of nephrotic syndrome in adults (Beck Jr et al., 2009; Ronco et al., 2021). Approximately one-third of patients show inadequate response to conventional therapy and may progress to renal failure within 5 to 10 years (Rovin et al., 2021). Although renal biopsy remains the gold standard for MN diagnosis, its invasiveness limits repeated clinical use. Currently, monitoring of MN and evaluation of therapeutic efficacy rely primarily on urinary protein and renal function measurements, which lack specificity. Therefore, identifying sensitive, specific, and noninvasive biomarkers is essential for diagnosing MN and tracking disease progression.

Currently, anti-PLA2R antibody is recognized as the classical marker for clinical MN diagnosis, but PLA2R-positive MN accounts for only 70%–80% of MN cases (Liu et al., 2022; Zhang et al., 2023). Investigating MN-associated cellular subpopulations and immune microenvironment molecular mechanisms may facilitate the discovery of novel molecular markers. Studies have indicated that altered T-cell proportions in MN patients linked to increased IgG4 levels, plasmacytosis, and reduced immune tolerance. Additionally, B-cell renal infiltration and intraglomerular complement system activation have been implicated in MN-related kidney injury (Rosenzwajg et al., 2017; Seifert et al., 2023). However, B-cells and T-cells constitute only a minor proportion of renal inflammatory cells in MN, which is predominantly driven by macrophages (Alexopoulos, Leontsini & Papadimitriou, 1994; Cai et al., 2024). The proportion of macrophages in urine correlates with patient response to treatment, while macrophage migration inhibitory factor (MIF) levels associate with MN disease activity and complications (Ding et al., 2022; Liu et al., 2023). Nevertheless, research on the characteristics and mechanisms of macrophages in MN kidney tissue remains limited. Additionally, glomerular endothelial cells, which directly interact with immune components in the bloodstream, have not yet been thoroughly investigated for their alterations and pathogenic mechanisms in MN.

This study employed transcriptomics, single-cell analysis, Mendelian randomization (MR), and clinical serum testing to explore the roles of macrophages and endothelial cells in MN. Notably, SLC1A5 protein was identified as a potential key molecular marker involved in the pathophysiology of MN. Differentially expressed genes associated with MN were revealed through bulk-transcriptome analysis of glomeruli, highlighting distinct macrophage subtypes in MN patients compared to healthy controls. Single-cell analysis demonstrated that these genes were primarily enriched in macrophages and endothelial cells, and time-trajectory analysis mapped their evolutionary dynamics. High-dimensional weighted gene co-expression network analysis (hdWGCNA) identified key genes driving cellular evolution, while MR established a causal link between SLC1A5 and MN pathogenesis. Finally, serum analysis further confirmed the correlation between SLC1A5 protein expression and disease progression in MN patients.

Methods and Materials

Transcriptome data acquisition

Glomerular microarray data from patients with MN and healthy transplanted kidneys were obtained from GEO database. The expression matrix was downloaded from the GSE104948 dataset, including samples from European Renal cDNA Bank subjects and living donors (Grayson et al., 2018). A total of 21 MN patient samples and 21 healthy donor samples were selected for subsequent analyses. Using GEO-provided annotation files, the expression matrix underwent Probe ID-Gene Symbol conversion. Additionally, single-cell RNA sequencing (scRNA-seq) data of kidney samples from MN patients (n =6) and healthy control subjects (n =2) were obtained from GSE171458 (Xu et al., 2021).

Identification of differentially expressed genes

The microarray transcriptome data were normalized and analyzed for differential expression using the limma R package (version 3.56.2). P-values were adjusted via Benjamini–Hochberg correction to generate adjusted p-values (p.adj). Genes with absolute log2-fold change (—log2FC—) > 1 and p.adj < 0.05 were identified as differentially expressed genes (DEGs). Differential analysis results were visualized using the ggplot2 and ComplexHeatmap of R packages.

Gene enrichment analysis

DEGs from microarray data underwent enrichment analysis using the KOBAS web tool (Xie et al., 2011). For scRNA-seq pseudotime and Olink proteomic analyses, gene lists were processed with the R package clusterProfiler (version 4.8.3) (Wu et al., 2021). Enrichment terms were derived from Kyoto Encyclopedia of Genes and Genomes (KEGG), Gene Ontology (GO), Reactome, and WikiPathways (WP) databases (Li et al., 2025; Xu et al., 2024). R package fgsea (version 1.26.0) was employed for gene set enrichment analysis (GSEA) on scRNA-seq data. The gene set pertaining to ferroptosis drivers was acquired from the FerrDb V2 database (Zhou et al., 2023).

Immune infiltration analysis

The CIBERSORT deconvolution algorithm was applied to the normalized glomerular expression matrix to estimate the proportions of various immune cells in tissues of both healthy and MN individuals (Newman et al., 2015). Specifically, LM22 feature matrix and 1,000 permutations were utilized for prediction, with quantile normalization performed. Box plots were constructed to display the proportions of each immune cell type in MN and healthy groups for the retained samples, and their inter-group differences were compared using Wilcoxon test. Spearman correlation coefficient was used to evaluate the relationship between expression of specific cytokines and immune infiltration.

scRNA-seq analysis

Data preprocessing, dimensionality reduction and clustering

ScRNA-seq profiles from six MN patients and two healthy donors were analyzed using the R package Seurat (version 4.3.0) (Butler et al., 2018). The filtered data underwent log-normalized, and variable genes were identified via the “FindVariableFeatures” function with the “vst” method. Log-normalized expression values were scaled, and clustering was performed using the top 50 principal components (PCs), with batch effects corrected by the R package Harmony (version 0.1.1). Cells were visualized in a two-dimensional plane using Uniform Manifold Approximation and Projection (UMAP), and cell cluster markers were identified using “FindAllMarkers” (Wu et al., 2024). Cell types were annotated by integrating SingleR package and markers from Cellmarker 2.0 and PanglaoDB (Franzén, Gan & Björkegren, 2019; Hu et al., 2023). Average gene expression across different cell types was calculated and visualized using the R package Heatmap.

Differential analysis of cell type abundance

We employed the miloR package (version 1.9.1) for the differential analysis of cell type abundance in single-cell data (Dann et al., 2022). Following the provided guidelines, a kNN graph was generated using the “buildGraph” function, and neighborhood was defined via the “makeNhoods” function. Differential analysis was then conducted to compare the variations in cell type abundance between cohorts with or without substantial massive proteinuria.

Pseudotime trajectory analysis

Pseudotime trajectories for monocytes/macrophages and endothelial cells were constructed separately using R package monocle (version 2.28.0) (Trapnell et al., 2014). Genes with mean expression ≥ 0.1 were selected via the “DispersionTable” function. Cells were mapped to a tree structure using DDRTree, with the starting point determined by cell differentiation scores from CytoTRACE (version 0.3.3), differential abundance analysis, and biological context. The “differentialGeneTest” function identified variable genes driving cell state changes over pseudotime, while the Branch Expression Analysis Modeling (BEAM) algorithm pinpointed genes dynamically altered during cell fate differentiation.

Cell communication analysis

Cell interactions within single-cell transcriptomes were examined using CellChat (version 1.6.1) (Jin et al., 2021).

High-dimensional weighted gene co-expression network analysis (hdWGCNA)

hdWGCNA was performed using the hdWGCNA R package (version 0.2.19), constructing Metacell objects with cell-type-specific parameters: monocytes/macrophages used kNN with k = 5, max shared cells = 10, min cells = 10, and soft power = 12; endothelial cells used k = 25 and soft power = 5. The analysis workflow included co-expression network construction, gene module clustering, eigengene-based connectivity (kME) calculation, and harmonized module eigengenes (hME) derivation. Module associations with clinical features were evaluated using the “ModuleTraitCorrelation” function.

Bidirectional mendelian randomization (MR)

To investigate the causal relationship between module gene expression and MN, we conducted a bidirectional Mendelian Randomization (MR) analysis using the TwoSampleMR package (version 0.5.7). eQTL data were obtained from the IEU OpenGWAS database and NephQTL (Gillies et al., 2018). We performed clumping to select suitable instrumental variables, excluding SNPs in linkage disequilibrium (r2 > 0.001) or lacking genome-wide significance (p ≥ 5 × 10−8). Instrument strength was assessed using the F-statistic, with F > 10 indicating robust associations. MN outcome data were sourced from the IEU OpenGWAS database, integrating datasets from diverse populations, including ebi-a-GCST010005 (2,150 cases and 5,829 controls of European ancestry) and ebi-a-GCST010004 (1,632 cases and 3,209 controls of East Asian ancestry) for discovery and validation. We applied the inverse-variance weighted (IVW) method for causal assessment, conducted heterogeneity testing to confirm IVW applicability, and assessed pleiotropy with the MR-Egger intercept test. Finally, reverse MR analysis was performed to explore reverse causality between gene expression and MN.

Serum SLC1A5 protein expression was detected by enzyme-linked immunosorbent assay (ELISA)

Serum samples from 50 MN patients and 31 healthy controls were collected from the First Affiliated Hospital of Soochow University. Serum was diluted 5-fold for analysis. Optical density (OD) was measured at 450 nm using a microplate reader, following the manufacturer’s instructions (Jingmei). SLC1A5 concentrations were quantified via a standard curve, with a detection limit of 1.0–160 pg/mL. The 50 MN patients were divided into high and low urinary protein groups, based on whether 24-hour urinary protein exceeded 3.5 g/day. Patients were further classified into nephrotic syndrome (urinary protein >3.5 g/day and serum albumin <30 g/L) and non-nephrotic syndrome groups. This study was retrospective, and the waived consent was given to and approved by the Medical Ethics Committee of the First Affiliated Hospital of Soochow University (2024-394).

Statistics

The aforementioned bioinformatics analyses were conducted using R 4.3.1 (R Core Team, 2023). Specific testing methods and p-value correction approaches were detailed in their respective methodology sections. The Wilcoxon rank-sum test was used to analyze the differences between two groups, and correlations were evaluated via Spearman’s test. A p-value of less than 0.05 was considered statistically significant.

Results

Key interactions between macrophages and endothelial cells in MN

In Fig. 1, we depict the key interactions between macrophages and endothelial cells in MN. Differential gene expression analysis revealed significant transcriptional variations in the glomeruli of MN patients compared to healthy donors. Subsequent single-cell localization studies indicated that these differentially expressed genes are predominantly enriched in macrophages and endothelial cells. Trajectory analysis and cell communication assessments identified pro-inflammatory macrophage and inflammatory endothelial cell subclusters, elucidating their critical interactions as central drivers of MN pathogenesis. Additionally, weighted gene co-expression network analysis (WGCNA) facilitated the identification of gene modules that are essential for each subcluster. Notably, Mendelian randomization (MR) analysis provided compelling evidence for a causal relationship between the module gene SLC1A5 and MN. Clinical validation further confirmed the significance of serum SLC1A5 protein expression, highlighting its potential as a biomarker for MN diagnosis and disease progression monitoring.

Figure 1 The key macrophage-endothelial cell interactions and potential regulatory genes in membranous nephropathy (MN).

Differential gene expression analysis identified genes upregulated in the glomeruli of MN patients compared to healthy donors, with single-cell localization revealing their enrichment in macrophages and endothelial cells. Trajectory analysis and cell communication highlighted the interactions of pro-inflammatory macrophage and inflammatory endothelial cell subclusters as critical in MN pathogenesis. Through weighted gene co-expression network analysis (WGCNA), essential gene modules associated with these subclusters were identified, while Mendelian randomization (MR) provided evidence of a causal relationship between the gene SLC1A5 and MN. Furthermore, the clinical relevance of serum SLC1A5 protein expression emphasizes its potential as a valuable diagnostic and prognostic biomarker in MN.Image credit: Servier Medical Art (https://smart.servier.com/), licensed under CC BY 4.0 (https://creativecommons.org/licenses/by/4.0/).

Bulk-transcriptome analysis of glomeruli revealed macrophage subtype differences between MN and control groups

The pathological changes of MN mainly affect the glomeruli, indicating that gene expression profiling in this region may elucidate its pathogenesis (Sealfon et al., 2022). Hence, gene expression analyses were performed on glomeruli from 21 MN patients and 21 healthy donors, identifying 95 differentially expressed genes (DEGs, —log2FC— > 1, p.adj < 0.05, Table S1). Of these, 39 genes were up-regulated, while 56 were down-regulated (Figs. 2A–2B).

Figure 2 Bulk transcriptomic analysis and immune infiltration of MN glomeruli.

(A) Volcano map showing the differentially expressed genes between MN and normal glomeruli. (B) Heatmap showing the relative expression of differentially expressed genes in different samples, labeled with the biological pathways they involved in. (C) Pathways differentially expressed genes significantly enriched in. (D) The CIBERSORT-based estimation of immune cell proportions in individual samples. (E) Boxplot illustrating the differences in immune infiltration between MN and control groups. (n =42). * 0.01 < p < 0.05, ** 0.001 < p < 0.01, *** p < 0.001, MN vs. Control, Wilcoxon test. MN, membranous nephropathy; Ctrl, control.

Subsequently, enrichment analysis was conducted to characterize the potential functions of differentially expressed genes. The results showed that up-regulated genes were primarily involved in immune-related pathways, including “Inflammatory response”, “Cytokine-cytokine receptor interaction”, “Chemokine signaling pathway” and “Neutrophil degranulation”. Conversely, down-regulated genes were significantly enriched in metabolic pathways, such as “Oxidation–reduction process”, “Metabolism of lipids”, “Glycolysis/Gluconeogenesis” and “PPAR signaling pathway” (Figs. 2B–2C).

The CIBERSORT algorithm was employed to estimate the proportion of various immune cell types in glomerulus and investigate changes in the immune microenvironment of MN glomerulus. The results revealed a significant increase in monocyte infiltration, as well as a notable rise in M1 macrophages (p < 0.05) and activated NK cells (p < 0.01), accompanied by a significant decrease in M2 macrophages (p < 0.001, Figs. 1D and 1E). Additionally, significant reductions were observed in naïve B-cells, neutrophils, and activated dendritic cells, suggesting that the immune milieu of MN glomerulus is uniquely altered, with particularly increased macrophage infiltration and disrupted polarization.

Expression patterns of up-regulated DEGs from transcriptome analysis in MN renal tissues were examined at the single-cell level

To elucidate the role of up-regulated DEGs in the MN immune microenvironment identified by transcriptome analysis, scRNA-Seq data from renal tissues of MN patients and healthy controls were reanalyzed. Following quality control and the reduction of batch effects using the R package harmony (Figs. S1A, S1B), a total of 30,671 cells were obtained. Ten distinct cell populations were initially identified through automated annotation with ingleR and prior knowledge, including proximal tubular cells, mesangial cells, podocytes, Loop of Henle cells, distal tubular cells, intercalated cells, principal cells, endothelial cells, fibroblasts/pericytes, and myeloid cells (Fig. 3A). The markers for these cell populations are displayed in Figs. 3B and 3C.

Figure 3 Dimensionality reduction and pseudotime analysis of scRNA seq data.

(A, B) UMAP visualization of cellular subtypes in renal tissues and expression distribution of PECAM1 and PTPRC. (C) Violin plot showing the expression of specific cell markers across different clusters. (D) Pseudotime development trajectory of macrophages. (E) Abundance changes of macrophages in different states between patients with massive proteinuria and those without. (F) Gene expression patterns and associated pathways of macrophages differentiating into different branches. (G) Pseudotime development trajectory of endothelial cells. (E) Abundance changes of endothelial cells in different states between patients with massive proteinuria and those without. (F) Gene expression patterns and associated pathways of endothelial cells alongside pseudotime. PT, proximal tubular cells; MC, mesangial cells; Pod, podocytes; LOH, Loop of Henle cells; DT, distal tubular cells; IC, intercalated cells; PC, principal cells; Endo, endothelial cells; Fib/Per, fibroblasts/pericytes; MP, massive proteinuria; noMP, no massive proteinuria.

Subsequently, the expression patterns of the 39 up-regulated DEGs identified by transcriptome analysis were explored at the single-cell level in MN renal tissues. These genes were found to be primarily localized in endothelial cells, fibroblasts/pericytes, myeloid cells, and podocytes (Fig. S1C). Given the low cell counts of fibroblasts/pericytes (383 cells) and podocytes (51 cells), the analysis focused on gene expression differences in myeloid and endothelial cells. Among 1,405 myeloid cells, 939 were identified as monocytes/macrophages, while other cell types, such as mixed lymphocytes and other myeloid cells, were excluded from further analysis due to insufficient numbers.

Pseudotemporal trajectory highlights pro-inflammatory macrophages and endothelial cells in NM progression

Monocle was used to create pseudotemporal trajectories for monocytes/macrophages and endothelial cells, tracking changes in cell states. For monocytes/macrophages, the trajectory revealed nine distinct states, forming two branches (Fig. 3D). Among them, State 6 exhibited the lowest differentiation potential, while State 9 was identified as the cluster with the highest differentiation potential (Figs. S1D, S1E). Both states were associated with patients presenting with significant proteinuria, although no such notable changes were observed between individuals with and without MN (Fig. 3E).

The Branching Expression Analysis Model (BEAM) identified genes undergoing significant changes during different cell state transitions. In the trajectory leading to branch 1 (State 6), inflammatory signaling pathways and proinflammatory genes (e.g., IL1B and CXCL8) were upregulated. Conversely, in the trajectory leading to branch 2, pathways related to the negative regulation of immune system and markers of M2 macrophages (e.g., CD163 and MRC1) were upregulated (Fig. 3F). Therefore, State 6 and State 9 were classified as proinflammatory macrophages and undifferentiated monocytes, respectively. These findings suggest that the immune environment within MN kidney is characterized by robust monocyte recruitment and differentiation into proinflammatory macrophages, consistent with transcriptome-based immune infiltration analyses.

Notably, the increased expression of VEGFA and significant enrichment of VEGFA-VEGFR2 signaling in proinflammatory macrophages suggest their potential involvement in the interaction between immune cells and glomerular capillaries.

For endothelial cells, monocle identified five states along a trajectory (Fig. 3G). Differential abundance analysis revealed a significant increase in State4 and decrease in State3 within MN patient kidneys (Fig. 3H). Cytotrace designated State4 as the cluster with the highest differentiation potential, while State3 exhibited the lowest (Figs. S1F, S1G). Although Cytotrace positioned State4 early in the differentiation sequence, due to its close association with disease progression and significant up-regulation of cytokines such as CXCL2 and CCL2 during the transition of cells towards State4 (Fig. 3I), it was considered as the endpoint of the pseudotime trajectory and classified as inflammatory endothelial cells (IECs). Furthermore, enrichment analysis indicated significant up-regulation of biological process such as “TNF signaling pathway”, “IL-17 signaling pathway”, “NF-kappa B signaling pathway” and “Autophagy” in IECs. Simultaneously, pathways including “Oxidative phosphorylation”, “ATP biosynthetic process”, “Signaling by VEGF” and “Glomerular filtration” were significantly down-regulated (Fig. 3J).

Cellular communication between pro-inflammatory macrophages and endothelial cells during MN progression

The strength of cell signaling pathways between distinct macrophage and endothelial cell (EC) subpopulations, particularly the cellular communication between pro-inflammatory macrophages and inflammatory endothelial cells (IECs), was characterized using CellChat. Compared to other macrophage subtypes, proinflammatory macrophages exhibited fewer autocrine ligand–receptor pairs but enhanced interaction with IECs. Conversely, IECs showed more autocrine ligand–receptor pairs than normal endothelial cells (Fig. 4A). At the signaling pathway level, the CCL signaling pathway was primarily secreted by proinflammatory macrophages and targeted anti-inflammatory macrophages, while also partially mediating autocrine effects in IECs. The CXCL signaling pathway predominantly facilitated interactions between various macrophage subtypes and IECs, whereas the VEGF signaling pathway was exclusively mediated by proinflammatory macrophages (Fig. 4B). At the molecular level, CXCL8 released by proinflammatory macrophages was shown to target IECs, while VEGFA acted on both endothelial cell types. Notably, normal endothelial cells exhibited higher sensitivity to VEGFA secreted by proinflammatory macrophages, possibly due to lower VEGFR expression in IECs. Additionally, autocrine effects in IECs were largely driven by CXCL2 and CCL2 (Fig. 4C). To further investigate the association between these cytokines and immune cells, Spearman correlation coefficients were calculated between cytokines expressions and immune cell proportions estimated by the CIBERSORT algorithm from RNA-seq data.

Figure 4 Interactions between endothelial cells and macrophages revealed by CellChat analysis.

(A) Number of interactions between cell subtypes. (B) Chord diagram illustrating signaling pathway communication between different cell subtypes. (C) Dotplot illustrating the interaction possibilities of ligand–receptor pairs between different cell subtypes. (D) Scatter plot demonstrating the correlation between specific cytokine expression and immune infiltration. Endo_inflam, inflammatory endothelial cell; endo_norm, normal endothelial cell; mac_anti, anti-inflammatory macrophage; mac_orig, undifferentiated monocyte; mac_pro, pro-inflammatory macrophage.

CXCL8 and CCL2 displayed significant positive correlations with monocyte infiltration, while CXCL8 was negatively correlated with M2 macrophages. Monocytes were positively correlated with M1 macrophages and negatively correlated with M2 macrophages (Fig. 4D). These findings suggest that CCL2 and CXCL8 may play a role in recruiting monocyte to the kidney and promoting their differentiation into M1 pro-inflammatory macrophages.

High-dimensional weighted gene co-expression network analysis of pro-inflammatory macrophages and endothelial cell-related key gene sets

Through transcriptome and single-cell analyses, pro-inflammatory cell subtypes and cellular communication closely related to MN pathogenesis and progression were identified. Subsequently, a set of genes potentially involved in regulating these cell subtypes was clustered using high-dimensional weighted gene co-expression network analysis (hdWGCNA). In monocytes/macrophages, 13 co-expression modules (Macro-M1 to Macro-M13) were identified and marked with distinct colors (Fig. 5A). Notably, Macro-M3 showed a significant positive correlation with pro-inflammatory macrophage phenotypes, while exhibiting negative correlations with undifferentiated monocytes and anti-inflammatory macrophages (Fig. 5B). Moreover, gene expression within this module was predominantly localized to State6 cells, corresponding to pro-inflammatory macrophages (Fig. 5C). In endothelial cells, six co-expression modules (Endo-M1 to Endo-M6) were identified (Fig. 5D). Among these, Endo-M2 and Endo-M4 exhibited significant positive correlations with the IEC phenotype, with gene expression concentrated in State4 cells, representing IECs (Figs. 5E and 5F). Figures 6A and 6C present the top 10 genes within each module. Functional enrichment analysis revealed common pathways in the key modules of pro-inflammatory macrophages and IECs, including ‘lipid and atherosclerosis’, ‘NF-kappa B signaling pathway’, ‘TNF signaling pathway’, and ‘apoptosis’. Furthermore, Macro-M3 was associated with ‘IL-17 signaling pathway’ and ‘antigen processing and presentation’, Endo-M2 with ‘NOD-like receptor signaling pathway’ and ‘HIF-1 signaling pathway’, while Endo-M4 was enriched in ‘other glycan degradation’, ‘ATP-dependent chromatin remodeling’, ‘sphingolipid metabolism’, and ‘peroxisome’ (Figs. 6B, 6D, and 6E).

Figure 5 Identification and selection of co-expressed modules in macrophages and endothelial cells.

(A) Dendrogram of macrophage genes clustered on the basis of topological overlap matrix. Each branch in the clustering tree represents a gene, while co-expression modules were constructed in different colors. (B) Module-trait heatmap of the correlation between macrophage modules and pseudotime states. (C) Distribution of co-expressed modules on macrophage UMAP plot represented by hME values and mapping of cellular pseudotime states. (D) Dendrogram of endothelial genes clustered on the basis of topological overlap matrix. Each branch in the clustering tree represents a gene, while co-expression modules were constructed in different colors. (E) Module-trait heatmap of the correlation between endothelial modules and pseudotime states. (F) Distribution of co-expressed modules on endothelial UMAP plot represented by hME values and mapping of cellular pseudotime states. * 0.01 < p < 0.05, ** 0.001 < p < 0.01, *** p < 0.001. Endo_inflam, inflammatory endothelial cell. hME, harmonized module eigengenes.

Figure 6 Functional enrichment analysis of genes in key modules.

(A) Top genes in macrophage-related module sorted by kME values. (B) Enriched functions of genes in the Macro-M3 module illustrated by a circos plot, with hME value of indicated modules in the right panel. (C) Top genes in endothelial-related module sorted by kME values. (D) Enriched functions of genes in the Endo-M2 and Endo-M4 module illustrated by a circos plot, with hME value of indicated modules in the right panel. kME, eigengene connectivity; hME, harmonized module eigengenes.

Bidirectional MR identified SLC1A5 as a potential causative gene for MN

To determine the causal relationship between key module gene expression and MN occurrence, bidirectional MR was conducted using eQTL data and GWAS summary data from two distinct MN cohorts with varying pedigrees. In the European ancestry discovery cohort, potential associations with MN were identified for 17 genes (Fig. 7A). The directional trend of OR values for these candidate genes was observed to persist in the East Asian pedigree validation cohort. However, only SLC1A5 (OR = 1.41, p = 0.02 in the discovery cohort; OR = 1.36, p = 0.02 in the validation cohort) and TRIM4 (OR = 1.21, p = 0.02 in the discovery cohort; OR = 1.15, p = 0.04 in the validation cohort) maintained statistical significance (Fig. 7B). The evaluation of heterogeneity and pleiotropy for the two genes, using Q-value and MR-Egger intercept, showed no significance (p > 0.05). This supported the use of the IVW approach and suggested a direct causal association between the exposure and outcome. In both cohorts, reverse MR analysis confirmed that gene expression levels caused MN and ruled out reverse causation (Fig. S3).

Figure 7 Bidirectional mendelian randomization unveiling potential causal genes of MN in key modules.

(A, B) Forest plot illustrating results of mendelian randomization in the discovery and validation cohort. (C) UMAP visualization of SLC1A5 expression in endothelial cells. (D) Gene set enrichment analysis of ferroptosis driver genes between SLC1A5 high and low expression groups. (E) Expression levels of indicated genes in SLC1A5 high and low expression groups (Wilcoxon test). MR, mendelian randomization; nSNP, number of single nucleotide polymorphism; Exp, expression; NES, normalized enrichment score.

Since SLC1A5 and TRIM4 were derived from the module associated with IECs, their expression in endothelial cells was examined. SLC1A5 expression was significantly concentrated in State4, representing IECs (Fig. 7C), while no such pattern was observed for TRIM4 (Fig. S4). As a key gene linked to ferroptosis, SLC1A5 was hypothesized to contribute to MN pathogenesis through endothelial cell ferroptosis. To investigate this, endothelial cells were stratified into high/low SLC1A5 expression groups, followed by GSEA. Endothelial cells with high SLC1A5 expression showed a notable upregulation of ferroptosis driver genes (Fig. 7D). Additionally, cytokine expression levels, including CCL2 and CXCL2, were elevated in SLC1A5-high-expressing endothelial cells (Fig. 7E).

SLC1A5 protein was highly expressed in the serum of MN patients and strongly associated with urinary protein levels

To evaluate the clinical relevance of SLC1A5 as a serum marker and its association with disease severity in MN, SLC1A5 protein expression was analyzed in 81 serum samples (31 healthy controls vs. 50 MN patients). The results revealed that SLC1A5 was significantly elevated in the serum of MN patients (p < 0.001) (Fig. 8A), and positively correlated with 24-hour total urinary protein (R = 0.418, p = 0.002) (Fig. 8B). Patients were divided into two groups based on urinary protein levels (>3.5 g/day vs. ≤3.5 g/day). In the high proteinuria group, SLC1A5 levels were significantly elevated in the serum (p < 0.001) (Fig. 8C). Further classification into nephrotic syndrome (NS) (urinary protein > 3.5 g/day + serum albumin <30 g/L) and non-NS groups revealed elevated SLC1A5 expression in the NS group (p = 0.047) (Fig. 8D).

Figure 8 SLC1A5 protein levels are elevated in the serum of MN patients and correlate strongly with urinary protein levels.

(A) SLC1A5 was significantly elevated in the sera of MN patients (p < 0.001). Indicating its potential as a secondary diagnostic marker for MN. (B) SLC1A5 protein expression in MN patients was positively correlated with 24-hour total urinary protein (R = 0.418, p = 0.002). (C) Serum SLC1A5 levels were significantly elevated in the high protein group (p < 0.001). (D) Serum SLC1A5 expression was elevated in the nephrotic syndrome group (p = 0.047).

Discussion

In this study, glomerular bulk transcriptome analysis identified 95 DEGs between MN and control groups. Immune infiltration analysis indicated macrophage infiltration and polarization were crucial in MN. Single-cell RNA sequencing further localized these DEGs to monocytes/macrophages and endothelial cells, highlighting their central roles in MN progression. Pseudotemporal trajectory analysis revealed shifts in the inflammatory states of these two cell types, and hdWGCNA technology identified molecular modules associated with these inflammatory cells. Notably, significant intercellular communication was observed, which hold important implications for MN.

Using MR with European and East Asian datasets, we confirmed causal associations between genes in these modules and MN risk. Among them, SLC1A5—a gene derived from endothelial cells—emerged as a key driver of MN progression. Validation in 81 clinical serum samples demonstrated that elevated SLC1A5 protein levels significantly correlated with urinary protein excretion, suggesting SLC1A5 as a potential therapeutic target for MN.

Transcriptome analysis revealed that upregulated genes in MN were primarily enriched in pathways related to chemokines, cytokines, neutrophil degranulation, and the innate immune system. Notably, the chemokine MCP-1 (CCL2) was highly expressed in renal tubular epithelial cells of patients with severe proteinuria and progressive MN, correlating with mesangial monocyte aggregation, consistent with immune infiltration analyses (Huber et al., 2002; Luo et al., 2021; Sun et al., 2021). Moreover, cytokines may exacerbate kidney injury by binding to CCR and CXCR chemokine receptors on podocytes, inducing apoptosis or facilitating the formation of tertiary lymphoid organs (Huber et al., 2002; Luo et al., 2021; Sun et al., 2021; Wang et al., 2022). Conversely, genes downregulated in MN glomerular tissues were enriched in pathways related to glycolysis, lipid metabolism, and oxidation–reduction processes, indicating metabolic dysregulation in affected kidneys. Wang et al. (2022) highlighted that mitochondrial dysfunction-induced podocyte proptosis may represent a potential mechanism in MN. Furthermore, antioxidants and oxygen radical scavengers have demonstrated exert positive effects in MN animal models (Liu et al., 2019). Collectively, these transcriptomic findings highlight the coexistence of intense inflammation and metabolic dysregulation in MN glomeruli, offering crucial insights into the disease’s underlying mechanisms.

Single-cell studies revealed that transcriptomic DEGs were predominantly concentrated in endothelial cells, macrophages, and podocytes, prompting a reassessment of the critical roles of endothelial cells and macrophages in the mechanisms of MN. Pseudotemporal trajectories and cell communication analyses indicated a strong association between pro-inflammatory macrophage subtypes, IEC subtypes, and severe proteinuria in MN patients. Specifically, pro-inflammatory macrophages highly expressed IL1B, CXCL8, and VEGFA, while IECs showed elevated CCL2 and CXCL2 expression. Elevated urinary CXCL8 levels correlate with poor prognosis in MN, and treatment with CXCL8/IL-8-blocking antibodies effectively reduced glomerular neutrophil aggregation and proteinuria (Chen et al., 2018). Notably, pro-inflammatory macrophages highly expressed VEGFA, a critical factor for podocyte and endothelial cell survival. VEGFA deficiency is associated with glomerular basement membrane thickening and barrier dysfunction, while its overexpression causes massive podocyte loss and severe proteinuria (Eremina et al., 2003; Tufro & Veron, 2012; Veron et al., 2010). Changes in VEGFA levels have been shown to induce subtle morphological alterations in glomerular endothelial cells, such as reduced migration and defects in the filtration barrier (Ursu et al., 2022). Cellular communication analysis revealed that IECs exhibited weaker interactions with VEGFA than normal endothelial cells. Furthermore, trajectory analysis demonstrated a progressive downregulation of endothelial development, cytoskeletal regulation, and glomerular filtration as cells transitioned to IECs, suggesting their significant role in the continued progression of proteinuria in MN. Although VEGF expression was relatively low in podocytes, its role in MN requires further investigation. CXCL2 and CCL2, secreted by IECs, have the ability to attract monocytes and promote their infiltration into local tissues, potentially shaping the renal immune microenvironment in MN. It was noted that the autocrine CCL2/CXCL2-ACKR1 axis in IECs maintained local chemokine homeostasis during inflammation (Torphy et al., 2022). These findings suggest that CXCL8, VEGFA, CCL2, and CXCL2 may serve as potential immunotherapeutic targets for MN.

HdWGCNA analysis identified key molecular modules in pro-inflammatory macrophages and endothelial cells, whose causal associations with MN pathogenesis were confirmed through MR methods. The study pinpointed the endothelial cell-derived gene SLC1A5 as a critical factor in MN progression, potentially promoting ferroptosis via excessive glutamine catabolism and insufficient glutathione synthesis. Notably, the ferroptosis pathway was significantly upregulated in endothelial cells with high SLC1A5 expression. Previous studies have demonstrated interactions between ferroptosis and inflammatory signaling pathways. Chen et al. (2022) reported that the NF-κB signaling pathway is activated during ferroptosis in smooth muscle cells, promoting the release of pro-inflammatory cytokines such as TNF, CXCL1, CXCL8, and CSF2. This suggests that the upregulation of CCL2 and CXCL2 in endothelial cells may be governed by similar regulatory mechanisms. In conclusion, SLC1A5-mediated endothelial ferroptosis drives cytokine release and morphological alterations in the glomerular filtration barrier, potentially exposing podocyte antigens and facilitating autoantibody deposition, which constitute the key events of MN pathogenesis. Elevated serum SLC1A5 protein expressions in MN patients correlate significantly with urinary protein levels, establishing its utility as a monitor for disease progression.

In our study, enzyme-linked immunosorbent assay (ELISA) showed a marked increase of circulating SLC1A5 in MN, whereas scRNA-seq comparing endothelial cells from MN and controls revealed only a modest, non-significant mRNA increase. This transcript–protein discordance can be explained by post-transcriptional regulation and membrane stabilization during endothelial activation. Activated/stressed endothelial cells also upregulate EV release, allowing SLC1A5 to enter the circulation independently of large mRNA changes. However, it is critical to note that SLC1A5 expression is not kidney-specific. Publicly available expression data from the GTEx database and Human Protein Atlas indicate substantial baseline expression of SLC1A5 in multiple extra-renal tissues, including the lung, skeletal muscle, large intestine, testis, T-cells, brain, and adipose tissue (Kanai et al., 2013; Poffenberger & Jones, 2014). Activated macrophages and other immune cells also upregulate SLC1A5 during metabolic reprogramming for immune responses (Nachef et al., 2021). While circulating SLC1A5 likely has multiple tissue sources, its positive correlation with proteinuria and inflammatory endothelial signatures supports a meaningful endothelial contribution in active MN. Thus, serum SLC1A5 concentrations likely reflect a combination of renal and extra-renal contributions, cellular expression changes, and tissue-specific shedding or secretion mechanisms. Further studies using animal models and comparative analysis of urinary and serum SLC1A5 levels will be necessary to clarify these complex origins.

This exploratory study has several limitations. First, functional validation of key genes such as SLC1A5 via cellular models and animal experiments is currently lacking, which hinders a comprehensive understanding of their mechanistic roles in MN pathogenesis. Second, although eQTL data from blood and kidney tissues were incorporated, the absence of single-cell level eQTL data from kidney tissues may compromise the precision of MR inferences. Future research should focus on the pathogenic mechanisms of pro-inflammatory macrophages and IECs in MN, as well as determining whether SLC1A5 regulates IECs differentiation through the ferroptosis pathway. These investigations are expected to enhance the understanding of MN’s molecular basis and facilitate the identification of effective therapeutic targets.

Conclusion

In conclusion, single-cell studies revealed that differentially expressed genes in MN glomeruli were predominantly located in monocytes/macrophages and endothelial cells, underscoring their critical role in MN development. Pseudotime and cellular communication analyses further revealed that CXCL8 from pro-inflammatory macrophages, along with CCL2 and CXCL2 secreted by IECs, may enhance renal monocyte infiltration. To investigate the mechanisms behind IECs, hdWGCNA and MR analyses were innovatively combined, which clearly indicated SLC1A5-mediated endothelial ferroptosis as a key mechanism in MN progression. Serum assays confirmed elevated SLC1A5 levels in MN patients, which positively correlated with urinary protein excretion. These findings offer novel insights into the molecular mechanisms of MN and nominate SLC1A5 as a potential therapeutic target.

Supplemental Information

Supplemental Information 1 STROBE-MR checklist of recommended items to address in reports of Mendelian randomization studies

Supplemental Information 2 The source code of Mendelian randomization analysis

Supplemental Information 3 Raw data of Figure 8A

Supplemental Information 4 Raw data of Figure 8B

Supplemental Information 5 Raw data of Figure 8C

Supplemental Information 6 Raw data of Figure 8D

Supplemental Information 7 Supplementary figures

Supplemental Information 8 The list of 95 differentially expressed genes

Supplemental Information 9 The basic characteristics of MN patients and healthy controls

We thank all the families for participating in this study.

Additional Information and Declarations

Competing Interests

Author Contributions

Human Ethics

Data Availability

The authors declare there are no competing interests.

Qingqin Tang performed the experiments, authored or reviewed drafts of the article, and approved the final draft.

Yi Ling performed the experiments, prepared figures and/or tables, and approved the final draft.

Jianzhong Li conceived and designed the experiments, prepared figures and/or tables, and approved the final draft.

Bin Feng performed the experiments, authored or reviewed drafts of the article, and approved the final draft.

Sheng Zhang performed the experiments, authored or reviewed drafts of the article, and approved the final draft.

Deyu Xu analyzed the data, prepared figures and/or tables, and approved the final draft.

Ling Zhou analyzed the data, prepared figures and/or tables, and approved the final draft.

Lei Shen conceived and designed the experiments, prepared figures and/or tables, and approved the final draft.

Guoyuan Lu conceived and designed the experiments, authored or reviewed drafts of the article, and approved the final draft.

Mingyu Chen analyzed the data, authored or reviewed drafts of the article, and approved the final draft.

Longwei Qiao performed the experiments, prepared figures and/or tables, and approved the final draft.

Yuting Liang conceived and designed the experiments, authored or reviewed drafts of the article, and approved the final draft.

The following information was supplied relating to ethical approvals (i.e., approving body and any reference numbers):

The institutional review board of the First Affiliated Hospital of Soochow University approval to carry out the study within its facilities (Ethical Application Ref: 2024-394).

The following information was supplied regarding data availability:

The code and raw data are available in the Supplemental Files.

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
