# Peer review of "Elevated SLC1A5 links to inflamed endothelial cells and proteinuria in membranous nephropathy patients"

_PeerJ, doi:10.7717/peerj.20271_

## Round 0.1 · original submission · Major Revisions

**Language Note:** PeerJ staff have identified that the English language needs to be improved. When you prepare your next revision, please either (i) have a colleague who is proficient in English and familiar with the subject matter review your manuscript, or (ii) contact a professional editing service to review your manuscript. PeerJ can provide language editing services - you can contact us at [email protected] for pricing (be sure to provide your manuscript number and title). – PeerJ Staff

Reviewer 1 ·

Basic reporting

The authors utilized published datasets and performed bioinformatic analysis. They identified altered genes and the enriched pathways in patients with MN compared to healthy controls. They then used pseudotemporal analysis on a publicly available single-cell dataset and observed an increase in proinflammatory macrophages and inflammatory endothelial cells. Using Mendelian randomization, the authors suggested SLC1A5 as a key gene in MN progression. They further conducted ELISA analysis in samples from 50 patients with MN and 31 healthy controls, and reported a high level of SLC1A5 in patients with MN, and attempted to demonstrate that SLC1A5 is a diagnostic and monitoring biomarker.
The manuscript is clearly written. Relevant prior literature is properly referenced. Some results/data need to be provided, for example, the list of 95 differentially expressed genes is important to be included as a supplementary table. Since the comprehensive bioinformatics analysis concluded on elevated SLC1A5 as a prognostic marker for MN, I have a few questions focusing on SLC1A5.

Experimental design

• Identifying a novel biomarker for early diagnosis and prognosis of MN is an unmet need. The study is well-designed. I have only one suggestion/question:
• The authors used the ERCB cohort to identify 95 genes that are differentially expressed in patients with MN compared to healthy controls. Since this gene list is the foundation of the entire study, a validation cohort to support the validity of this gene list is critical. In the same paper where the ERCB MN data were extracted, there is also NEPTUNE cohort transcriptomic data for MN patients available. Validation using NEPTUNE MN patients will strengthen this paper.

Validity of the findings

• Correlation method: Figure 4D: considering the distribution of the data and the presence of outliers (particularly for the CXCL8 correlations), Spearman correlation is more appropriate to use than Pearson in this case. The same applies to Figure 8 B.

• Is SLC1A5 expressed more in glomeruli or tubulointerstitium? Is it one of the 95 genes originally identified using the glomerular dataset? How is it regulated in the ERCB compared with healthy controls? Do you expect higher SLC1A5 mRNA in patients with MN?

• The vendor of the SLC1A5 ELISA is not provided. This is key information that needs to be included in the methods. Also, provide the following information to allow reviewers to evaluate the validity of this ELISA data: were samples measured in duplicate or single measurement? What is the inter- and intra-plate CV of the measurement? The detection limit of 0 is unusual. Please provide the lower limit of quantification.

• The difference between healthy controls and patients with MN is dramatically different; if this can be validated in independent cohorts, SLC1A5 indeed may be a good diagnostic biomarker. However, for diagnostic purposes, the authors also need to show that SLC1A5 is not present in samples from another nephrotic syndrome (FSGS and MCD) and proteinuric (IgAN) patients. Interestingly, the SLC15A level does not strongly correlate with 24-hour UPRO. How about its correlation with GFR? For better interpretation of the SLC1A5 with clinical outcome, please provide a table with the basic characteristics of the 50 MN patients and the 31 controls, including age, sex, GFR, PCR, etc.

• It is important to discuss the source of the serum/plasma SLC1A5 in the discussion, considering both renal and extra-renal origins. Within the kidney, identifying the specific cell types that predominantly secrete or shed SLC1A5 is crucial. Please provide a dot plot analysis showing SLC1A5 expression across different kidney cell types, stratified by disease and control groups, which would be preferred.

Additional comments

'MN" needs to be spelled out in the article title.
Page 16, line 300: Macro-M13 might be a typo
Page 15, line 280: typo: "NM" should be "MN"

Reviewer 2 ·

Basic reporting

This manuscript is well written and well presented.

Experimental design

Well done on the experimental design

Validity of the findings

The findings seem valid

Additional comments

The authors do a good job of finding SLCA15 as a marker related to endothelial cells/macrophages, but their claim of this being used as a diagnostic marker is quite immature (without any prospective sensitivity/specificity experiments). The authors should try to tone down this claim in the manuscript and present it for what it is.
The authors must also clearly state in detail why they chose to focus on SLC1A5 contribution to MN (and if other genes were considered, but didn't fit a given certain criteria).
The authors also do not talk about the roles of SLC1A5 in MN and validate with any animal model. A couple of in vitro experiments in this regard, along with references, would really help the impact of the manuscript.

---

## Round 0.2 · Major Revisions

Reviewer 1 ·

Basic reporting

No comment in addition to my previous review.

Experimental design

No comment.

Validity of the findings

1. The SLC1A5 ELISA data were crucial to a key conclusion. However, the samples in this study were measured in single measurement of each sample using an ELISA assay that I could not locate through an online search. Please provide the link to the vendor's website, along with the assay feasibility and validation data, which should be available from the vendor. It is important to conduct a matrix effect and stability analysis of serum samples to ensure the assay's quality (such information should be available from the vendor). Typically, with a relatively small sample size, ELISA data should be presented in duplicates for each sample to ensure robust and reliable data, allowing for the calculation of the coefficient of variation (CV).
2. In the authors' response letter, Table S2 indicates that the mean concentration of SLC1A5 in healthy controls is significantly higher than in patients with MN, which contradicts Figure 8A. Please double-check if the labeling and the data is correct.
3. Is there a link between the high circulating SLC1A5 level in MN v.s Healthy control and the modest and insignificantly increased SLC1A5 RNA expression in endothelial cells? if so, what is the underlying mechanism? This needs to be discussed.

---

## Round 0.3 · accepted · Accept

Thank you for revising your manuscript to address the reviewers' concerns. Reviewer 1 now recommends acceptance and I am satisfied that the comments of Reviewer 2 have been addressed. The manuscript is now ready for publication.

Reviewer 1 ·

Basic reporting

No additional comments in addition to my previous review

Experimental design

No additional comments in addition to my previous review

Validity of the findings

The authors addressed my concerns properly.

Additional comments

No additional comments.